# CLIP-ViP: Adapting Pre-trained Image-Text Model to Video-Language Alignment

**Hongwei Xue**[1,*] **Yuchong Sun**[2,*] **Bei Liu**[3,†] **Jianlong Fu**[3,†]
**Ruihua Song**[2], **Houqiang Li**[1], **Jiebo Luo**[4]
[1]University of Science and Technology of China, Hefei, China,
[2]Renmin University of China, Beijing, China,
[3]Microsoft Research, Beijing, China,
[4]University of Rochester, Rochester, NY

## Abstract

Pre-trained image-text models, like CLIP, have demonstrated the strong power of vision-language representation learned from a large scale of web-collected image-text data. In light of the well-learned visual features, there are works that transfer image representation to the video domain and achieve good results. However, adapting image-text pre-trained models to video-text pre-training (i.e., post-pretraining) has not demonstrated a significant advantage yet. In this paper, we tackle this challenge by raising and addressing two questions: 1) what are the factors hindering post-pretraining CLIP from improving performance on video-text tasks, and 2) how to mitigate the impact of these factors. Through a series of comparative experiments and analyses, we find that the data scale and domain gap between language sources have large impacts. By these observations, we propose an Omnisource Cross-modal Learning method equipped with a **Vi**deo **P**roxy mechanism on the basis of CLIP, namely CLIP-ViP. Extensive results show that our approach improves the performance of CLIP on video-text retrieval by a large margin. Our model achieves state-of-the-art results on a variety of datasets, including MSR-VTT, DiDeMo, LSMDC, and ActivityNet. We release our code and pre-trained CLIP-ViP models at `https://github.com/microsoft/XPretrain/tree/main/CLIP-ViP`.

## 1 Introduction

In the past few years, vision-language pre-training has achieved great success on cross-modal representation learning from a large scale of web-crawled data (Radford et al., 2021; Jia et al., 2021; Li et al., 2021; Wang et al., 2021b; Zellers et al., 2021; 2022; Bain et al., 2021). Among them, image-text pre-trained models (Radford et al., 2021; Jia et al., 2021) have shown powerful capability for various downstream tasks, including visual understanding (Gu et al., 2021; Wang et al., 2021a; Rao et al., 2022) , image-text generation (Patashnik et al., 2021; Mokady et al., 2021) and so on (Guzhov et al., 2022; Zhang et al., 2022). In light of the well-learned and enriched visual representation, some works directly adapt image-text pre-trained models to video-text downstream tasks without further pre-training on video data (Luo et al., 2021; Fang et al., 2021; Gorti et al., 2022; Zhao et al., 2022), while still outperforming models pre-trained on video data (Xu et al., 2021b; Bain et al., 2021).

Utilizing an existing powerful image-text pre-trained model for further video-text pre-training (i.e., post-pretraining) is able to reduce the required training cost by making good use of the knowledge learned from images. However, adapting image-text pre-trained models to video-text data for post-pretraining has not demonstrated a significant advantage yet, and thus is still under-explored. A preliminary study is conducted by CLIP4Clip (Luo et al., 2021) which adopts MeanPooling by averaging multiple frame features based on the CLIP model on a subset of Howto100M (Miech et al., 2019). While the improvement over directly using the image-text pre-trained model is marginal for

---

*Equal contributon. This work was performed when Hongwei Xue and Yuchong Sun were visiting Microsoft Research as research interns.

†Corresponding authors.

either zero-shot or fine-tuning settings. In this paper, we aim to explore how to effectively adapt the image-text pre-trained model (e.g., CLIP) to video-language representation learning for video-text tasks (e.g., text-to-video retrieval).

To unleash the power of video data to adapt image-text pre-trained models for post-pretraining, we conduct several preliminary experiments to figure out the challenges that hinder post-pretraining. First, we explore post-pretraining an image-text pre-trained model (i.e., CLIP) with MeanPooling on video-text datasets with different scales, including WebVid-2.5M (Bain et al., 2021) and HD-VILA-100M (Xue et al., 2022). The result shows that the scale of data is critical for video-text post-pretraining. Data on a small scale makes the model easy to over-fit the new data while the knowledge learned from image-text is suppressed and the performance is reduced. Second, we investigate the language domain gap between pre-training data and downstream data. By calculating the Normalized Mutual Information (NMI) on clusters of text features, we find that there is a large domain gap between subtitles that are used in large-scale video-text pre-training data and descriptive texts in downstream tasks.

To mitigate the impact of the above factors, we propose CLIP-ViP to adapt the pre-trained image-text model CLIP for video-text pre-training. First, we introduce auxiliary captions that have a smaller language domain gap with downstream data into existing large-scale video-text data. Instead of using a video captioning model which may cause data leakage by training on the same dataset with video-text downstream tasks, and considering a better visual captioning capability, we adopt an image captioning model to generate an auxiliary caption of middle frame in each video. In order to adapt a Transformer-based vision encoder to process both images and videos with minimal modification, we then propose video proxy tokens and design a proxy-guided video attention mechanism for the Vision Transformer (ViT). Specifically, during attention computation in each block, video proxy tokens can interact with all tokens, while patch tokens only interact with video proxy tokens and patch tokens within the same frame. Our vision encoder only increases negligible parameters and calculations compared to the vanilla Vision Transformer while increasing the generality and extendability. To facilitate cross-modal representation learning from both caption-frame and video-subtitle data types at the same time, we propose an Omnisource Cross-modal Learning (OCL) method for pre-training and study a series of variants to find the best fusion strategy.

Our experimental results show that our approach improves the performance of CLIP on text-to-video retrieval tasks by a large margin. We also conduct ablation studies to verify the effectiveness of each part in our approach. Our contributions are summarized as follows: (1) We are one of the first to explore factors that hinder video post-pretraining on pre-trained image-text models; (2) We propose CLIP-ViP that can effectively leverage image-text pre-trained model for post-pretraining; (3) We conduct extensive experiments to verify the effectiveness of our method. Our model outperforms the state-of-the-art results by a large margin on four widely-used benchmarks.

## 2 RELATED WORK

**Vision-Language Pre-Training** End-to-end models (Lei et al., 2021; Xue et al., 2022; Zellers et al., 2021; Fu et al., 2021; Huang et al., 2020; 2021b; Xue et al., 2021; Li et al., 2021; Kim et al., 2021; Huang et al., 2021a; Sun et al., 2022) for vision-language pre-training are replacing the traditional approach using pre-extracted visual features by off-the-shelf models (Sun et al., 2019; Xu et al., 2021b; Zhu & Yang, 2020; Li et al., 2020b;a; Chen et al., 2020). Training end-to-end models on large-scale web-collected data also gradually demonstrates the big advantages (Radford et al., 2021; Jia et al., 2021; Xue et al., 2022; Zellers et al., 2021; 2022). Unlike images that have alt-texts, large-scale video datasets suitable for pre-training usually use subtitles as text sources (Miech et al., 2019; Xue et al., 2022). Subtitles are much noisier than alt-texts, according to (Miech et al., 2019), typical examples of incoherence include the content producer asking viewers to subscribe to their channel, talking about something unrelated to the video, or describing something before or after it happens. Bain *et al.* collect a video dataset WebVid (Bain et al., 2021) with textual description annotations. Their texts are well aligned with the video and avoid suffering from ASR errors. However, the vast majority of WebVid videos are sourced from a stock footage website, so scaling up is under limitation. The video-subtitle data is more easily accessible on the web and thus suitable for scaling up. In this paper, we investigate the unfavorable factors of video-subtitle data and explore how to mitigate the impact of these factors.

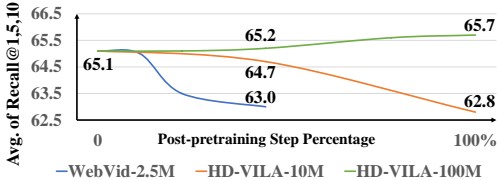

| NMI Score | MSR-VTT | DiDeMo | Mean |
|---|---|---|---|
| HD-VILA$_{sub}$ | 0.831 | 0.684 | 0.758 |
| HD-VILA$_{cap}$ | 0.317 | 0.621 | 0.469 |
| WebVid | 0.420 | 0.488 | 0.454 |
| COCO | 0.373 | 0.758 | 0.566 |
| CC12M | 0.445 | 0.673 | 0.559 |

Figure 1: The curve of finetuning results during post-pretraining. The x-axis indicates the percentage of training steps. The y-axis indicates average value of Recall@1, 5 and 10. [Best viewed in color]

Table 1: Normalized Mutual Information (NMI) score of language features extracted on series of data and downstream tasks. Larger value indicates larger domain gap.

**Pre-trained Models for Video-Text Retrieval**  The great success of the CLIP has demonstrated its unprecedented power on varies downstream tasks, including vision understanding (Gu et al., 2021; Wang et al., 2021a; Rao et al., 2022), image-text generation (Patashnik et al., 2021; Mokady et al., 2021) and so on (Guzhov et al., 2022; Zhang et al., 2022). By contrastive learning on large-scale image-text pairs, CLIP learns enriched visual concepts for images. Recently, some works directly transfer CLIP to video-text retrieval without further pretraining on video data (post-pretraining) (Luo et al., 2021; Fang et al., 2021; Gorti et al., 2022; Zhao et al., 2022; Wang et al., 2022c). Their work takes the performance of video-text retrieval to a new level, outperforming existing models pre-trained on video data (Xu et al., 2021b; Bain et al., 2021; Xue et al., 2022; Ge et al., 2022; Wang et al., 2022a). They transfer CLIP from views of feature aggregation (Luo et al., 2021; Zhao et al., 2022; Fang et al., 2021; Gorti et al., 2022) or representation alignment (Fang et al., 2021; Gorti et al., 2022; Wang et al., 2022c). In parallel with these works, we study post-pretraining with video data on top of CLIP in an effective way and our model can be combined with other approaches effectively.

## 3  PRELIMINARY ANALYSIS

In this section, we first study the impact of the data scale for adapting image-text pre-training to video-text post-pretraining, and then investigate how the language domain gap affects the adaption.

### 3.1  POST-PRETRAINING WITH DIFFERENT DATA SCALES

To study the effectiveness of different data scales, we use the CLIP-ViT-B/32 model (Radford et al., 2021) as the base image-text pre-trained model and adopt MeanPooling for video adaption like CLIP4Clip (Luo et al., 2021) by averaging multiple frame features as video feature. Two open-domain video-text datasets are used: WebVid-2.5M (Bain et al., 2021) with 2.5 million pairs and HD-VILA-100M (Xue et al., 2022) with 100M pairs. We also adopt a subset of HD-VILA-100M containing random 10% data (namely HD-VILA-10M) as a middle setting. We run the same number of steps on all settings, equivalent to one epoch on HD-VILA-100M. We uniformly sample 12 frames from each video and apply the same hyper-parameters as described in Section 5 for all settings.

During post-pretraining, we evaluate the pre-trained models by fine-tuning on MSR-VTT text-to-video retrieval task. Figure 1 shows the performance trend. We observe an overfitting phenomenon that continuous post-pretraining leads to a performance drop. And the drop is more significant for smaller data (e.g., WebVid-2.5M and HD-VILA-10M). As CLIP is pre-trained on 400 million image-text pairs, further training on small data makes the model tend to overfit the new data while the implicit knowledge learned from the image-text pairs is degrading. As a consequence, the performance will drop, even worse than using CLIP directly. Thus we adopt HD-VILA-100M due to its large scale and diverse category.

### 3.2  LANGUAGE DOMAIN GAP WITH DOWNSTREAM DATA

It is intuitive that pre-training on data with the same domain as downstream data can benefit downstream tasks. For most video-text tasks like video-text retrieval, texts are descriptive sentences of videos (i.e., captions). While for HD-VILA-100M, which we will use for pre-training, the texts are auto-transcribed subtitles and they indicate very different relevance to visual information compared to descriptive texts. Meanwhile, auto-transcribed subtitles suffer from irrelevance, misalignment, and ASR errors (Miech et al., 2019). To better explore the language domain gap between pre-

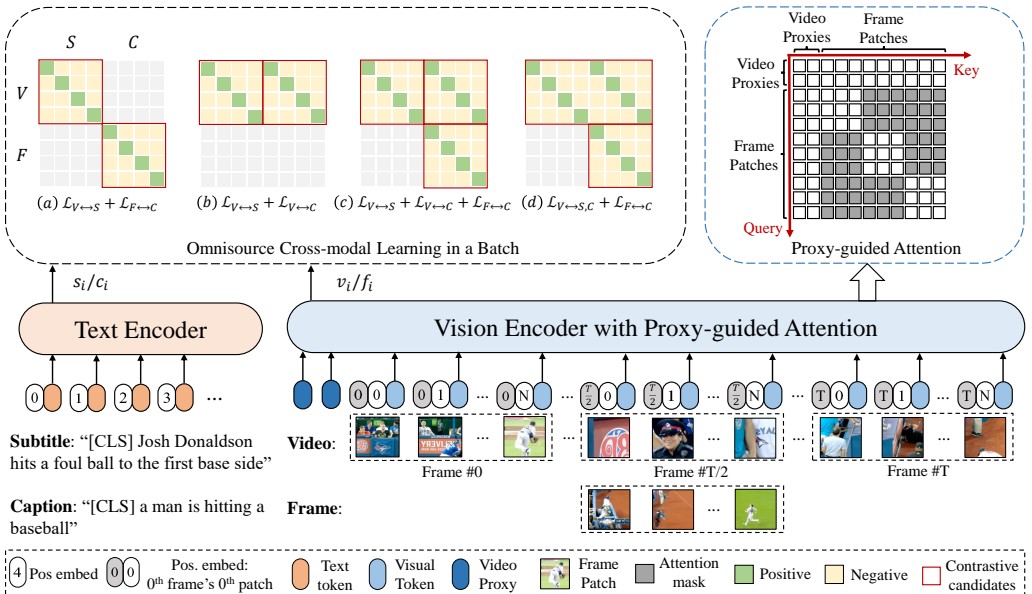

Figure 2: The framework of CLIP-ViP with a text encoder and a vision encoder. Taken features $V$, $F$, $S$, $C$ of videos, frames, subtitles, captions as input, a series of Omnisource cross-modal learning variants are studied to explore better representation learning losses: (a) $\mathcal{L}_{V \leftrightarrow S} + \mathcal{L}_{F \leftrightarrow C}$; (b) $\mathcal{L}_{V \leftrightarrow S} + \mathcal{L}_{V \leftrightarrow C}$; (c) $\mathcal{L}_{V \leftrightarrow S} + \mathcal{L}_{V \leftrightarrow C} + \mathcal{L}_{F \leftrightarrow C}$; (d) $\mathcal{L}_{V \leftrightarrow S,C} + \mathcal{L}_{F \leftrightarrow C}$. In the vision encoder, Video proxy tokens and the ViP-guided attention mechanism is proposed to transfer CLIP into the video domain. [Best viewed in color]

training data and downstream data, we measure the inconsistency by calculating the dissimilarity between their language features. For downstream language data, we choose two typical video-text retrieval datasets: MSR-VTT (Xu et al., 2016) and DiDeMo (Anne Hendricks et al., 2017). For pre-training language, we select four types: video subtitles of HD-VILA-100M (HD-VILA$_{sub}$), video captions of WebVid-2.5M, image captions of MS-COCO (Lin et al., 2014), and web-collected alt-texts of Conceptual Caption 12M (Changpinyo et al., 2021). In addition, we analyze auto-generated captions of HD-VILA-100M (HD-VILA$_{cap}$), which will be introduced in Section 4.

We use a Transformer Encoder initialized from CLIP (Radford et al., 2021) to extract text features. To quantify the domain gap of languages between pre-training and downstream data, we first mix their text features and then use K-means to get two clusters. Then we calculate the Normalized Mutual Information (NMI) between cluster labels and ground-truth labels of pre-training or downstream. A larger NMI value means that the two types of features are easy to be distinguished, thus there is a larger domain gap. For each comparison, we randomly sample 1000 texts from each type of data for 10 times and adopt the average of 10 results. We report the results in Table 1. Comparing the values of all pre-training data types, we find that the NMI score between HD-VILA$_{sub}$ and downstream data is much larger than others, especially for MSR-VTT downstream dataset. This indicates that direct training with subtitles may introduce inconsistency with downstream tasks.

## 4 APPROACH

In this section, we will introduce the proposed CLIP-ViP video pre-training framework. To bridge language domain gaps between image and video datasets, we first introduce an in-domain auxiliary data generation method. Then, we propose a novel Video Proxy mechanism to enable the Vision Transformer (ViT) model for both image and video encoding. We further present an Omnisource Cross-modal Learning (OCL) method which can jointly learn cross-modal representation from video-text and image-text pairs.

### 4.1 IN-DOMAIN AUXILIARY DATA GENERATION

Motivated by the analysis in Section 3, we introduce auxiliary captions into large-scale video-subtitle data to reduce the language domain gap between pre-training and downstream data. We adopt an

image captioning model for two reasons. 1) Most SOTA video captioning models are trained with video-text datasets (e.g., MSR-VTT, ActivityNet) which are also used for downstream tasks. We avoid data leakage to perform pre-training agnostic to downstream data. 2) The performance of existing video captioning models lags far behind that of images. Thus, we choose a powerful image captioning model OFA-Caption (Wang et al., 2022b) to generate one caption for the middle frame of each video in HD-VILA-100M. We use the default setting of the OFA-Caption model. As a result, we generate 100M sentences with a max length of 16 words. This method can be applied to any video data and we will release the generated captions to facilitate future research.

## 4.2 VIDEO PROXY MECHANISM

Since video is an ordered sequence of frames, it is critical to learn the frame aggregation and temporality when transferring to the video domain. Meanwhile, to keep the high generality and extendability of the Vision Transformer (ViT) backbone, we aim to find a simple but effective way to transfer ViT to enable both image and video encoding with minimal modifications. Given a video containing $T$ frames: $\{f_1, f_2, ..., f_T\}$, we follow CLIP to divide each frame into N patches: $\{f_t^1, f_t^2, ..., f_t^N \mid t \in [1, T]\}$. Then we add spatio-temporal positional embedding to each flattened 2D patches:

$$g(f_t^n) = Linear(f_t^n) + Pos_s(n) + Pos_t(t),  \tag{1}$$

where $Linear(*)$ is a linear layer, $Pos_s(n)$ and $Pos_t(t)$ is the learnable spatial and temporal positional embedding, respectively. The whole video can be divided into $T \times N$ patch tokens.

To model spatial information from multi-frames, one simple way is directly feeding all tokens into CLIP's vision encoder and conducting attention across all tokens. However, this method introduces significant conflicts with CLIP. As CLIP is pre-trained on image and text pairs, it has difficulty handling interactions of tokens between images/frames during training. We also verify it by experiments as Full Attention setting in Table 2. Instead, we introduce a Video Proxy token to act as a proxy that helps each local patch perceive video-level temporal information.

Before feeding into CLIP, we concatenate patch tokens with a set of learnable parameters called video proxy tokens: $\mathcal{P} = \{p_1, p_2, ..., p_M\}$, where $M$ is the number of video proxy tokens. Then all $T \times N + M$ tokens will be fed into the ViT of CLIP. The output of the first video proxy token will be regarded as the video's representation. We also design a proxy-guided attention mechanism for the vanilla ViT. In the attention score calculation of each block, video proxy tokens attend to all tokens, while patch tokens only attend to tokens in the same frame plus video proxy tokens. This mechanism can be formulated as attention mask $\mathcal{M}_{\mathrm{ViP}}$:

$$\mathcal{M}_{\mathrm{ViP}}(q, k) = \begin{cases} 1 & \text{if } q \in \mathcal{P} \text{ or } k \in \mathcal{P} \text{ or } (q, k) \text{ in the same frame,} \\ 0 & \text{otherwise,} \end{cases}  \tag{2}$$

where $q$ and $k$ is the key and query in attention calculation. Patch tokens can obtain global information from video proxy tokens while reducing inconsistencies with the original CLIP's calculation. Our experiment in Section 5 demonstrates the superiority of this mechanism.

For the input type of the image/frame, we use linear interpolation to get a middle temporal positional embedding, then treat the image/frame as a special single-frame video. This method enables joint training on both videos and images in the same batch, as our proxy-guided attention mechanism reduces the difference in calculations between video and image.

## 4.3 OMNISOURCE CROSS-MODAL LEARNING

To learn rich video-language alignment from video-subtitle pairs and reduce the language domain gap with downstream data by corresponding auxiliary frame-caption pairs, we study joint Cross-Modal Learning on the omnisource input. Following most works of learning multimodal alignment on dual encoders (Radford et al., 2021; Xue et al., 2022; Li et al., 2021; Luo et al., 2020; Xu et al., 2021b; Luo et al., 2021), we use info-NCE loss to perform contrastive learning. There are two formats of visual source : video sequences and single frames, and two types of text source : subtitles and captions in our work. We denote them by $V$, $F$, $S$, and $C$ respectively. We define a source-wise info-NCE loss by:

$$\mathcal{L}_{v2t} = -\frac{1}{B} \sum_{i=1}^{B} \log \frac{e^{v_i^\top t_i / \tau}}{\sum_{j=1}^{B} e^{v_i^\top t_j / \tau}}, \quad \mathcal{L}_{t2v} = -\frac{1}{B} \sum_{i=1}^{B} \log \frac{e^{t_i^\top v_i / \tau}}{\sum_{j=1}^{B} e^{t_i^\top v_j / \tau}}  \tag{3}$$

| Model | R@1 ↑ | R@5 ↑ | R@10 ↑ | Mean ↑ |
|---|---|---|---|---|
| MeanPool | 43.4 | 70.9 | 81.1 | 65.1 |
| SeqTransformer | 44.6 | 71.2 | 81.8 | 65.9 |
| Full Attention | 42.8 | 70.1 | 80.3 | 64.4 |
| 2 Video Proxy Tokens | 45.8 | 71.3 | 81.7 | 66.3 |
| 4 Video Proxy Tokens | **46.5** | 72.1 | **82.5** | **67.0** |
| 8 Video Proxy Tokens | 45.7 | **72.7** | 81.7 | 66.7 |

Table 2: MSR-VTT text-to-video retrieval results of finetuning CLIP by different settings. Mean ↑ indicates the average value of Recall@1, 5, and 10. All results are based on CLIP-ViT-B/32.

| Model | MSR-VTT Retrieval | | | | DiDeMo Retrieval | | | |
|---|---|---|---|---|---|---|---|---|
| | R@1 ↑ | R@5 ↑ | R@10 ↑ | Mean ↑ | R@1 ↑ | R@5 ↑ | R@10 ↑ | Mean ↑ |
| CLIP-MeanPool | 43.4 | 70.9 | 81.1 | 65.1 | 40.6 | 67.5 | 77.2 | 61.8 |
| CLIP-ViP | 46.5 | 72.1 | 82.5 | 67.0 | 40.6 | 70.4 | 79.3 | 63.4 |
| $+ \mathcal{L}_{V \leftrightarrow S}$ | 47.7 | 72.1 | 82.4 | 67.4 | 44.6 | 73.9 | 81.9 | 66.8 |
| $+ \mathcal{L}_{V \leftrightarrow S} + \mathcal{L}_{F \leftrightarrow C}$ | 49.3 | 74.8 | 83.8 | 69.3 | 48.4 | 74.5 | 84.4 | 69.1 |
| $+ \mathcal{L}_{V \leftrightarrow S} + \mathcal{L}_{V \leftrightarrow C}$ | 49.6 | 74.2 | 84.0 | 69.3 | 48.5 | 76.6 | 83.6 | 69.5 |
| $+ \mathcal{L}_{V \leftrightarrow S} + \mathcal{L}_{V \leftrightarrow C} + \mathcal{L}_{F \leftrightarrow C}$ | 49.6 | 74.5 | 83.8 | 69.3 | 48.5 | 76.8 | 84.1 | 69.8 |
| $+ \mathcal{L}_{V \leftrightarrow S,C} + \mathcal{L}_{F \leftrightarrow C}$ | 49.6 | 74.5 | 84.8 | **69.6** | 48.2 | 76.7 | 84.4 | **69.8** |

Table 3: Ablation study of different losses. We report text-to-video results of models finetuned on MSR-VTT and DiDeMo. Mean ↑ means the average of Recall@1, 5 and 10. All results are based on CLIP-ViT-B/32. All post-pretraining steps are equivalent to one epoch on HD-VILA-100M.

where $v_i$ and $t_j$ are the normalized embeddings of $i$-th visual feature in $X \in \{V, F\}$ and $j$-th text feature in $Y \in \{S, C\}$ in a batch of size $B$. $\tau$ is a learnable temperature. The overall alignment loss $\mathcal{L}_{X \leftrightarrow Y}$ is the average of $\mathcal{L}_{v2t}$ and $\mathcal{L}_{t2v}$. For example, $\mathcal{L}_{V \leftrightarrow S}$ represents info-NCE loss within video-subtitle pairs in a batch, which pulls aligned pairs together in embedding space while pushing apart misaligned pairs.

We study the reasonable variants of OCL: **(a)** $\mathcal{L}_{V \leftrightarrow S} + \mathcal{L}_{F \leftrightarrow C}$: Simple combination of two source-wise losses on video-subtitle and frame-caption pairs; **(b)** $\mathcal{L}_{V \leftrightarrow S} + \mathcal{L}_{V \leftrightarrow C}$: As there is also content correlation between videos and its middle-frame captions, we explore to add a loss on video-caption pairs to baseline loss $\mathcal{L}_{V \leftrightarrow S}$; **(c)** $\mathcal{L}_{V \leftrightarrow S} + \mathcal{L}_{V \leftrightarrow C} + \mathcal{L}_{F \leftrightarrow C}$: Combination of (a) and (c); **(d)** $\mathcal{L}_{V \leftrightarrow S,C} + \mathcal{L}_{F \leftrightarrow C}$: A video corresponds to both a subtitle and auxiliary caption. Compare to (c), the numbers of negative pairs in $\mathcal{L}_{v2t}$ can be expanded. The $\mathcal{L}_{v2t}$ in $\mathcal{L}_{V \leftrightarrow S,C}$ is rewritten as:

$$\mathcal{L}_{v2t} = -\frac{1}{2B} \sum_{i=1}^{B} (\log \frac{e^{v_i^\top s_i / \tau}}{\sum_{j=1}^{B} e^{v_i^\top s_j / \tau} + e^{v_i^\top c_{j \neq i} / \tau}} + \log \frac{e^{v_i^\top c_i / \tau}}{\sum_{j=1}^{B} e^{v_i^\top c_j / \tau} + e^{v_i^\top s_{j \neq i} / \tau}}), \quad (4)$$

where $s_i \in S$ and $c_i \in C$. The $\mathcal{L}_{t2v}$ in $\mathcal{L}_{V \leftrightarrow S,C}$ is equal to **(c)**. We compare all variants with the baseline $\mathcal{L}_{V \leftrightarrow S}$ and report results in Section 5.

## 5 EXPERIMENT

### 5.1 EXPERIMENTAL DETAILS

**Video-Text Post-Pretraining.** To pre-train the proposed CLIP-ViP model, we uniformly sample 12 frames and resize all frames to 224×224 from video clips with an average length of 13.4 seconds. The sampled frames can well cover the semantics conveyed from a video clip. For text, we adopt the CLIP's tokenizer to split a sentence into word tokens with a max length of 70. We use AdamW optimizer (Loshchilov & Hutter, 2019), and empirically set an initial learning rate as 5e-6 and a fixed weight decay as 5e-2. For the learning rate schedule, we adopt a cosine decay with a warm-up strategy. We train our model with 32 NVIDIA Tesla V100 GPUs in a batch size of 1024. The contrastive similarity is calculated on gathered features from all GPUs. We set training steps to one epoch on HD-VILA-100M for all ablation studies and three epochs for the full setting.

**Fine-tuning Training.** To better adapt CLIP-ViP to downstream tasks, we reuse most hyper-parameters of post-pretraining in fine-tuning with some exceptions. 1) Batch size: we fine-tune

| Post-pretrain Data | MSR-VTT Retrieval | | | | DiDeMo Retrieval | | | |
|---|---|---|---|---|---|---|---|---|
| | R@1 ↑ | R@5 ↑ | R@10 ↑ | Mean ↑ | R@1 ↑ | R@5 ↑ | R@10 ↑ | Mean ↑ |
| w/o Post-pretrain | 46.5 | 72.1 | 82.5 | 67.0 | 40.6 | 70.4 | 79.3 | 63.4 |
| HD-VILA$_{sub}$ | 47.7 | 72.1 | 82.4 | 67.4 | 44.6 | 73.9 | 81.9 | 66.8 |
| HD-VILA$_{cap}$ | 45.9 | 73.0 | 81.8 | 66.9 | 44.9 | 74.4 | 82.3 | 67.2 |
| HD-VILA$_{sub+cap}$ | **49.6** | **74.5** | **84.8** | **69.6** | **48.2** | **76.7** | **84.4** | **69.8** |
| ImageCaption | 45.6 | 70.7 | 81.1 | 65.8 | 43.7 | 69.5 | 77.9 | 63.7 |
| HD-VILA$_{sub}$ + ImageCaption | 49.1 | 73.1 | 83.5 | 68.6 | 47.0 | 75.3 | 84.1 | 68.8 |

Table 4: Ablation study of post-pretrain data. We report text-to-video results of models finetuned on MSR-VTT and DiDeMo. Mean ↑ indicates an average of Recall@1, 5 and 10. For all results, the model is designed with 4 video proxy tokens and pre-trained on CLIP-ViT-B/32. All post-pretraining steps are equivalent to one epoch on HD-VILA-100M.

our model with a batch size of 128 for all downstream tasks for a fair comparison. 2) Learning rate and weight decay: we empirically set them to 1e-6 and 0.2, respectively. 3) Number of epochs: due to the various scales of downstream datasets, we set epoch numbers to 5, 20, 10, and 20 for MSR-VTT, DiDeMo, LSMDC, and ActivityNet, respectively. 4) Frame number: for a fair comparison, we set frame number to 12 except for ActivityNet Captions (set to 32) as its videos are much longer (180 seconds on average). Note that the hyper-parameters of downstream training are the same in all settings in the ablation study.

**Downstream Datasets.** To evaluate performances of video pre-training models, we conduct text-to-video retrieval experiments on four typical datasets. **(a) MSR-VTT** (Xu et al., 2016) contains 10K YouTube videos with 200K descriptions. We follow previous works (Yu et al., 2018; Liu et al., 2019) to train models on 9K videos, and report results on the 1K-A test set. **(b) DiDeMo** (Anne Hendricks et al., 2017) consists of 10K Flickr videos annotated with 40K sentences. We follow (Liu et al., 2019; Zhang et al., 2018) to evaluate paragraph-to-video retrieval and concatenate all descriptions of a video as one query. **(c) LSMDC** (Rohrbach et al., 2016) consists of 118,081 video clips sourced from 202 movies with one caption corresponding to each clip. Evaluation is conducted on a test set of 1,000 videos from movies disjoint from the train and validation sets. **(d) ActivityNet Captions** (Krishna et al., 2017a) contains 20K YouTube videos annotated with 100K sentences. We follow the paragraph-to-video retrieval setting (Zhang et al., 2018; Liu et al., 2019) to train models on 10K videos and report results on the val1 set with 4.9K videos.

## 5.2 ABLATION STUDIES

**Video Proxy Mechanism.** For the vision encoder, we evaluate our proposed Video Proxy (ViP) mechanism with different numbers of proxies and compare it with different model structures (i.e. MeanPool, SeqTransformer, Full Attention) by fine-tuning the same pre-trained model on MSR-VTT retrieval task. MeanPool simply takes the average of frame features as the representation of the whole video. For SeqTransformer, we follow the seqTransf type in CLIP4Clip (Luo et al., 2021) and the residual connection in their implementation. Full Attention setting takes all patch tokens as the input of the vision encoder and attention is conducted across all tokens. All models are initialized with CLIP-ViT-B/32. The results are shown in Table 2. Compared to the MeanPool baseline which completely disregards temporality, SeqTransformer improves the average Recall@1,5,10 by 0.8%. Full Attention type leads to a significant performance drop and we observe a worse initial status and slower convergence than other settings during the experiment. This is consistent with our analysis in Section 4.2 that directly using CLIP for all patches' attention computation will decrease the advantage of CLIP. In our method, different numbers of video proxy tokens all result in significant performance gain on R@1 (e.g., 3.1% by 4 proxies), while only increasing negligible parameters: 3K compared to 86M of ViT backbone. Compared with other settings, our methods in all settings have the most improvement which indicates that our proposed video proxy mechanism can effectively leverage the image-text pre-trained model for video-text post-pretraining.

**Omnisource Cross-modal Learning.** To verify the effectiveness of the proposed Omnisource Cross-modal Learning (OCL) and compare its variants, we set a post-pretraining and fine-tuning pipeline and adopt the same hyper-parameters for all experiments. $\mathcal{L}_{V \leftrightarrow S}$ is the baseline contrastive

| Method | MSR-VTT Retrieval | | | | ActivityNet Captions Retrieval | | | |
|---|---|---|---|---|---|---|---|---|
| | R@1 ↑ | R@5 ↑ | R@10 ↑ | Mean ↑ | R@1 ↑ | R@5 ↑ | R@10 ↑ | Mean ↑ |
| ClipBERT (Lei et al., 2021) | 22.0 | 46.8 | 59.9 | 6.0 | 21.3 | 49.0 | 63.5 | 6.0 |
| VLM (Xu et al., 2021a) | 28.1 | 55.5 | 67.4 | 4.0 | - | - | - | - |
| MMT (Gabeur et al., 2020) | 26.6 | 57.1 | 69.6 | 4.0 | 28.7 | 61.4 | - | 3.3 |
| Support Set (Patrick et al., 2021) | 30.1 | 58.5 | 69.3 | 3.0 | 29.2 | 61.6 | - | 3.0 |
| Frozen (Bain et al., 2021) | 31.0 | 59.5 | 70.5 | 3.0 | 28.8 | 60.9 | - | 3.0 |
| VideoCLIP (Xu et al., 2021b) | 30.9 | 55.4 | 66.8 | - | - | - | - | - |
| HD-VILA (Xue et al., 2022) | 35.6 | 65.3 | 78.0 | 3.0 | 28.5 | 57.4 | - | 4.0 |
| Florence (Yuan et al., 2021) | 37.6 | 63.8 | 72.6 | - | - | - | - | - |
| All-in-One (Wang et al., 2022a) | 37.9 | 68.1 | 77.1 | - | 22.4 | 53.7 | 67.7 | 5.0 |
| BridgeFormer (Ge et al., 2022) | 37.6 | 64.8 | 75.1 | 3.0 | - | - | - | - |
| *CLIP-ViT-B/32:* | | | | | | | | |
| CLIP4Clip (Luo et al., 2021) | 44.5 | 71.4 | 81.6 | 2.0 | 40.5 | 72.4 | - | 2.0 |
| CenterCLIP (Zhao et al., 2022) | 44.2 | 71.6 | 82.1 | 2.0 | 43.9 | 74.6 | 85.8 | 2.0 |
| XPool (Gorti et al., 2022) | 46.9 | 72.8 | 82.2 | 2.0 | - | - | - | - |
| CLIP2Video (Fang et al., 2021) | 45.6 | 72.6 | 81.7 | 2.0 | - | - | - | - |
| CLIP2Video†(Bogolin et al., 2022) | 47.2 | 73.0 | 83.0 | 2.0 | - | - | - | - |
| CLIP2TV (Gao et al., 2021) | 46.1 | 72.5 | 82.9 | 2.0 | 44.1 | 75.2 | - | 2.0 |
| DRL (Wang et al., 2022c) | 47.4 | 74.6 | 83.8 | 2.0 | 44.2 | 74.5 | 86.1 | 2.0 |
| CAMoE* (Cheng et al., 2021) | 47.3 | 74.2 | 84.5 | 2.0 | 51.0 | 77.7 | - | - |
| Ours | 50.1 | 74.8 | 84.6 | 1.0 | 51.1 | 78.4 | 88.3 | 1.0 |
| Ours* | **55.9** | **77.0** | **86.8** | 1.0 | **59.1** | **83.9** | **91.3** | 1.0 |
| *CLIP-ViT-B/16:* | | | | | | | | |
| CenterCLIP (Zhao et al., 2022) | 48.4 | 73.8 | 82.0 | 2.0 | 46.2 | 77.0 | 87.6 | 2.0 |
| CLIP2TV (Gao et al., 2021) | 49.3 | 74.7 | 83.6 | 2.0 | - | - | - | - |
| DRL (Wang et al., 2022c) | 50.2 | 76.5 | 84.7 | 1.0 | 46.2 | 77.3 | 88.2 | 2.0 |
| DRL†(Wang et al., 2022c) | 53.3 | 80.3 | 87.6 | 1.0 | - | - | - | - |
| Ours | 54.2 | 77.2 | 84.8 | 1.0 | 53.4 | 81.4 | 90.0 | 1.0 |
| Ours* | **57.7** | **80.5** | **88.2** | 1.0 | **61.4** | **85.7** | **92.6** | 1.0 |

Table 5: Comparison with SOTA models in MSR-VTT (Xu et al., 2016) and ActivityNet (Krishna et al., 2017a) text-to-video retrieval tasks. * and † respectively denotes that the method uses DSL (Cheng et al., 2021) and QB-Norm (Bogolin et al., 2022) as post-processing operations.

loss on video-subtitle pairs. After introducing auxiliary captions, we study four variants of OCL Loss: (a) $\mathcal{L}_{V\leftrightarrow S}+\mathcal{L}_{F\leftrightarrow C}$; (b) $\mathcal{L}_{V\leftrightarrow S}+\mathcal{L}_{V\leftrightarrow C}$; (c) $\mathcal{L}_{V\leftrightarrow S}+\mathcal{L}_{V\leftrightarrow C}+\mathcal{L}_{F\leftrightarrow C}$; (d) $\mathcal{L}_{V\leftrightarrow S,C}+\mathcal{L}_{F\leftrightarrow C}$ as explained in Section 4.3. We pre-train models with each loss function for only one epoch due to the costly training, then finetune on two video-text retrieval datasets: MSR-VTT and DiDeMo. We compare the results with CLIP-MeanPool and CLIP using the proposed Video Proxy mechanism without post-pretraining (i.e., CLIP-ViP). The results are listed in Table 3. On MSR-VTT dataset, we find that $\mathcal{L}_{V\leftrightarrow S}$ brings very little improvement: 0.4% on average of Recall@1, 5, 10. This is due to the large domain gap between MSR-VTT and post-pretraining data. Combined with auxiliary captions, four variants of OCL loss all bring significant improvements: over 3% on Recall@1 and over 2.3% on average of Recall@1, 5, 10. On DiDeMo dataset, based on the improvement brought by $\mathcal{L}_{V\leftrightarrow S}$, OCL further improve the results by a large margin: 8% on average of Recall@1, 5, 10. Finally, $\mathcal{L}_{V\leftrightarrow S,C}+\mathcal{L}_{F\leftrightarrow C}$ performs best which is applied as our final setting.

**Auxiliary Data.** In this part, we ablate the contribution of large-scale noisy data and auxiliary data. For uni-source, we use video-subtitle pairs and video-caption data for post-pretraining by vanilla contrastive loss. For data combination, we apply OCL under $\mathcal{L}_{V\leftrightarrow S,C}+\mathcal{L}_{F\leftrightarrow C}$ setting to post-pretrain on the combined data. From Table 4, Omnisource post-pretraining results are much better than two uni-source results. On MSR-VTT, both uni-source post-pretraining show limited improvement: 67.4% and 66.9% compared with 67.0%. While the Omnisource post-pretraining brings a significant improvement of 2.6%. On DiDeMo, the benefit of data combination is also obvious, with nearly double the improvements brought by uni-source. These results show that the auxiliary data together with our designed joint learning method can effectively adapt the image-text pre-trained model to video-text post-pretraining.

As the generation of auxiliary captions is based on OFA-Caption (Wang et al., 2022b), a powerful image-text pre-trained model, we also explore only including existing data in post-pretraining. We choose image-text pairs of several widely-adopted datasets: MS-COCO, Visual Genome (VG) (Krishna et al., 2017b), Flickr-30K (Young et al., 2014), SBU (Ordonez et al., 2011), CC3M (Sharma et al., 2018) and CC12M as our auxiliary data (namely ImageCaption). To ablate the contribution of these data, we add experiments of post-pretraining on ImageCaption alone and HD-VILA-100M

| Method | DiDeMo Retrieval | | | | LSMDC Retrieval | | | |
|---|---|---|---|---|---|---|---|---|
| | R@1↑ | R@5↑ | R@10↑ | Mean↑ | R@1↑ | R@5↑ | R@10↑ | Mean↑ |
| MMT (Gabeur et al., 2020) | - | - | - | - | 12.9 | 29.9 | 40.1 | 19.3 |
| ClipBERT (Lei et al., 2021) | 20.4 | 48.0 | 60.8 | 6.0 | - | - | - | - |
| Frozen (Bain et al., 2021) | 31.0 | 59.8 | 72.4 | 3.0 | 15.0 | 30.8 | 40.3 | 20.0 |
| HD-VILA (Xue et al., 2022) | 28.8 | 57.4 | 69.1 | 4.0 | 17.4 | 34.1 | 44.1 | 15.0 |
| All-in-One (Wang et al., 2022a) | 32.7 | 61.4 | 73.5 | 3.0 | - | - | - | - |
| BridgeFormer (Ge et al., 2022) | 37.0 | 62.2 | 73.9 | 3.0 | 17.9 | 35.4 | 44.5 | 15.0 |
| *CLIP-ViT-B/32:* | | | | | | | | |
| CLIP4Clip (Luo et al., 2021) | 43.4 | 70.2 | 80.6 | 2.0 | 21.6 | 41.8 | 49.8 | 11.0 |
| CenterCLIP (Zhao et al., 2022) | - | - | - | - | 21.7 | 39.8 | 49.8 | 11.0 |
| XPool (Gorti et al., 2022) | - | - | - | - | 22.7 | 42.6 | 51.2 | 10.0 |
| CLIP2TV (Gao et al., 2021) | 45.5 | 69.7 | 80.6 | 2.0 | - | - | - | - |
| DRL (Wang et al., 2022c) | 47.9 | 73.8 | 82.7 | 2.0 | 24.9 | 45.7 | **55.3** | 7.0 |
| CAMoE* (Cheng et al., 2021) | 43.8 | 71.4 | - | - | 25.9 | 46.1 | 53.7 | - |
| Ours | 48.6 | 77.1 | 84.4 | 2.0 | 25.6 | 45.3 | 54.4 | 8.0 |
| Ours* | **53.8** | **79.6** | **86.5** | 1.0 | **26.0** | **46.4** | 54.9 | 8.0 |
| *CLIP-ViT-B/16:* | | | | | | | | |
| CLIP4Clip by (Zhao et al., 2022) | - | - | - | - | 24.1 | 45.0 | 55.1 | 8.0 |
| CenterCLIP (Zhao et al., 2022) | - | - | - | - | 24.2 | 46.2 | 55.9 | 8.0 |
| DRL (Wang et al., 2022c) | 49.0 | 76.5 | 84.5 | 2.0 | 26.5 | 47.6 | 56.8 | 7.0 |
| Ours | 50.5 | 78.4 | 87.1 | 1.0 | 29.4 | 50.6 | 59.0 | 5.0 |
| Ours* | **55.3** | **82.0** | **89.3** | 1.0 | **30.7** | **51.4** | **60.6** | 5.0 |

Table 6: Comparison with SOTA models in DiDeMo (Anne Hendricks et al., 2017) and LSMDC (Rohrbach et al., 2016) text-to-video retrieval tasks. * denotes using post-processing DSL (Cheng et al., 2021).

combined with ImageCaption. From Table 4, post-pretraining on ImageCaption alone results in performance degradation on MSR-VTT and marginal improvement on DiDeMo. In contrast, Image-Caption yields significant performance gains on both datasets when used as auxiliary data for HD-VILA-100M. This further illustrates the importance of the combination of large-scale noisy data and auxiliary data.

## 5.3 Comparison to State-of-the-art Models

We compare our model under full setting (in three epochs) with state-of-the-art works on the text-to-video retrieval task. The results of fine-tuning on four datasets (i.e., MSR-VTT, DiDeMo, ActivityNet Captions, LSMDC) are shown in Table 5 and 6, respectively. We clarify the backbone for CLIP-based works. We only add results with DSL (Cheng et al., 2021) to make fair comparison with some methods using post-processing operations e.g., DSL (Cheng et al., 2021) or QB-Norm (Bogolin et al., 2022). Our model achieves the best results on all datasets in both CLIP-ViT-B/32 and CLIP-ViT-B/16. Note that some existing methods are also applicable on top of our models as our modification to the CLIP model is minimal. Note that even without post-processing (e.g., DSL), our results still surpass methods using post-processing operations on most datasets. Besides, adding DSL will greatly improve the performance of our model since our model has good bidirectional vision-language correspondence. The good results on the ActivityNet Captions dataset also indicate that our model can generalize well to long videos. Overall, the improvements on different datasets demonstrate the superiority of the video-language representation learned by our CLIP-ViP model.

## 6 Conclusion

In this paper, we study further pre-training (post-pretraining) image-text models like CLIP on large-scale video data. We first conduct a preliminary analysis to reveal the factors hindering video post-pretraining. Motivated by findings, we propose CLIP-ViP which includes an Omnisource Cross-modal Learning method and a Video Proxy mechanism. The Video Proxy mechanism can better model videos containing temporal information while reducing conflicts with the pre-trained CLIP model. The Omnisource Cross-modal Learning alleviates the problem caused by the domain gap between video-subtitle and downstream data. Extensive results show that our approach improves the performance of CLIP on video-text retrieval by a large margin and also achieves new state-of-the-art results on four widely-used video-language benchmarks.

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

## A    MORE DOWNSTREAM TASKS

We add various suitable downstream tasks to verify that our model can be generalizable to other tasks. Specifically, we add four zero-shot evaluations on video action recognition and video-language multi-choice tasks:

**HMDB51 Action Recognition**    HMDB51 (Kuehne et al., 2011) is a video dataset containing realistic videos from various sources, including movies and web videos. There are 51 action categories (such as "jump", "kiss" and "laugh"). We evaluate our model by zero-shot classification on HMDB51. To adapt to our model, we formulate video classification as video-text alignment, using text prompts "a person [label]". The prediction is made on top-1 similarity between query video and prompted texts. We directly test our model without finetuning to report zero-shot accuracy results.

**SomethingSomethingV2 (SSv2) Action Recognition.**    SomethingSomethingV2 (Goyal et al., 2017) dataset is a large collection of labeled video clips that show humans performing pre-defined basic actions with everyday objects. It requires models to develop fine-grained understanding of basic actions that occur in the physical world. To adapt to our model, we design a multi-choice task on SSv2. The model has to choose the ground truth annotation among all categories. We directly test our model without finetuning to report zero-shot accuracy results.

**MSR-VTT Multi-Choice (MC).**    MSR-VTT MC is a benchmark build on MSR-VTT video dataset. Given a video query and five descriptive sentences, MSR-VTT MC requires the model to choose a single best answer in the candidates. We directly test our model without finetuning to report zero-shot accuracy results.

**LSMDC Multi-Choice (MC).**    LSMDC MC is a benchmark build on movie data. Similar with MSR-VTT MC, it requires the model to choose a single best answer from 5 candidates. We report zero-shot accuracy on 10000 test videos.

From the results in Table 7, we can find that post pretraining CLIP on subtitle alone will lead to a performance drop on most tasks: HMDB51 SSv2 and MSR-VTT MC. This phenomenon is in consistency with our assumption as subtitles usually have unique form. By our final setting, CLIP-ViP

| Model | HMDB51 | SSv2 | MSR-VTT MC | LSMDC MC |
|---|---|---|---|---|
| CLIP-MeanPool | 40.3 | 25.8 | 89.9 | 64.3 |
| CLIP-ViP w/o post-pretraining | 41.1 | 26.1 | 89.4 | 63.5 |
| CLIP-ViP w/ video-subtitle | 36.6 | 28.0 | 86.6 | 67.0 |
| CLIP-ViP full | **44.8** | **33.0** | **90.3** | **67.6** |

Table 7: Evaluation on more downstream tasks. We report zero-shot accuracy (%) on HMDB51, SSv2, MSRVTT-MC, and LSMDC MC. All models are base size with patch size of 32. Details of post-pretraining are the same as in Section 5.2.

achieves better results than CLIP on all evaluations, especially on SSv2 which requires more understanding on temporality (Wang et al., 2022d). These results demonstrate that CLIP-ViP outperforms CLIP on various video benchmarks and further show the generalization of our model CLIP-ViP.

## B EFFICIENCY OF POST-PRETRAINING

In light of the good image representation of CLIP, our post-pretraining significantly reduces the training cost, Compared with HD-VILA (Xue et al., 2022) which trains video-language representation from scratch, CLIP-ViP outperforms HD-VILA by a large margin with about 1/30 training time. To make more fair comparison with training from scratch. We conduct an experiment of training from scratch with the same setting and training time (one epoch). Then we finetune on MSR-VTT Retrieval to evaluate the pre-trained model.

| Model | MSR-VTT R@1 | MSR-VTT R@5 | MSR-VTT R@10 |
|---|---|---|---|
| CLIP-ViP (from scratch) | 28.5 | 55.2 | 68.6 |
| CLIP-ViP | 49.6 | 74.5 | 84.8 |

Table 8: Comparison with training CLIP-ViP from scratch. We report finetuning results on MSR-VTT Retrieval. All models are base size with patch size of 32. Details of post-pretraining are the same as in Section 5.2.

From the results in Table 8, we can see that there is a huge performance gap. Without basing on a CLIP, CLIP-VIP can not leverage the rich knowledge entailed in CLIP, thus leading to a much lower training efficiency.

