# OpenReview forum: "CLIP-ViP: Adapting Pre-trained Image-Text Model to Video-Language Alignment"
_ICLR.cc/2023/Conference — ICLR 2023 poster_

### Official Review · Reviewer_XvUh · 2022-10-24

**Confidence:** 5
**Correctness:** 3
**Technical Novelty And Significance:** 2
**Empirical Novelty And Significance:** 2
**Recommendation:** 5

**Clarity, Quality, Novelty And Reproducibility:**

The writing of this paper is easy to read, and their proposed model shows good results on multiple datasets. The major concerns are the use of the powerful image captioning model and the contribution of model design and losses. The current major contribution is proxy-guide attention (the combination of contrastive loss is still limited).

**Strength And Weaknesses:**

- As for Further pretraining on small data, some details are missing; for example, did the authors try to freeze the backbone and optimize the head only?

- The adopted OFA was powerful and could generate diverse and well captions. However, such a strong captioning model also brings another problem, what if we do not use OFA, but use previous models such as BUTD, SCST, etc. Then what are the effects of image captions on the final performance?

- The proxy-guided attention looks new and interesting, and the results shown in Table 2 look promising. Showing some visualized attention samples will help the understanding of what the proxy learned.

- The cross-modal learning part is straightforward and combines several contrastive learning tasks. The novelty of this part is somehow limited, especially considering that in table 3, some losses do not bring a big performance. They simply combine some possible losses together and lack strong motivation.

- Given that they have already adopted OFA as the caption generation module and they have already got a large amount of data, then what if they train the model without the initial weight from CLIP? I think such a comparison can also show how much benefit their model leverages from the CLIP model.

**Summary Of The Paper:**

This paper starts with two questions that try to answer what and how to apply CLIP to video-text tasks. They build a model based on the observations and propose new cross-modal learning methods for video-text retrieval. Their method was tested and showed promising results on those datasets. Adapting the image-caption pretarined model to video is a challenge. The findings in this paper can serve as a good baseline for the bridge of image-caption pretrained models and vision-text tasks.

**Summary Of The Review:**

The proposed model achieved good performance on test datasets. The additional computation and proxy-guided attention need to be discussed more.

---

> ### Author Response · Authors · 2022-11-18
> **Response to Reviewer XvUh**
>
> We sincerely thank you for the constructive comments. We address your major concerns as below.
>
> **1. Experiments in Preliminary Analysis**
> For Fig 1's experiment, we try various hyper-parameters for training, including small learning rates for encoders or frozen encoders. The conclusion stays the same. It's worth noting that the "overfitting" here has the different meaning. Usually overfitting means the model fits too well on its training data, thus cannot perform accurately against unseen data. However, in our scenario, "overfitting" means further training on new data will cause degradation of the knowledge learned by CLIP. Due to the difference, this phenomenon cannot be easily solved by optimization tricks.
>
> **2. Image Captions**
> Since our target is to improve video-language post-pretraining on the basis of CLIP, the most powerful image captioning model would be our first choice. Intuitively, a weaker captioning model will lead to performance drop while we don't think there is necessity for this evaluation. Instead of looking back, we believe more powerful captioning models in the future will empower our work better. Also we would like to emphasize that CLIP-ViP mainly benefits from the combination of large-scale video-subtitle data and auxiliary data, as depicted in Tab 4.
>
> **3. Visualization Results**
> We directly inference our model and provide some visualized attention samples here.
> We put these samples in following links, Each figure contains 4 rows (number of proxy tokens) and 12 columns (number of frames).
> <https://i.imgur.com/sh4EE0E.jpg>
> <https://i.imgur.com/nGfCaFG.jpg>
> <https://i.imgur.com/eoOpK3r.jpg>
> <https://i.imgur.com/JqYTJQe.jpg>
> <https://i.imgur.com/j0hmKMb.jpg>
> <https://i.imgur.com/9hVGS18.jpg>
> <https://i.imgur.com/biqLv8G.jpg>
> <https://i.imgur.com/oVMXZsX.jpg>
> <https://i.imgur.com/fTicqKx.jpg>
> <https://i.imgur.com/GdDF4Lh.jpg>
> <https://i.imgur.com/0xCQLch.jpg>
>
> From these results, we find that the proxy tokens usually concentrate on multiple salient regions to aggregate key information in a video.
>
>
> **4. Study of Cross-Modal Loss**
> We emphasize the contribution of cross-modal learning part is that we are the first to explore how to utilize contrastive learning to **simultaneously process videos and images** in vision-language alignment learning.
> We **parallelly** study different variants of losses. According to Tab. 2, $L_{V \leftrightarrow S,C} + L_{F \leftrightarrow C}$ achieves the best results, benefiting from expanding negative samples.
>
> **5. Training from Scratch**
> In light of the good image-language representation learned by CLIP, our post-pretraining significantly reduces the training cost. Compared to HD-VILA which trains video-language representation from scratch, **CLIP-ViP outperforms HD-VILA by a large margin (refer to Tab 5,6) with about 1/30 training time**. To make more fair comparison with training from scratch, we add an experiment setting to pre-train from scratch with the same setting and training time (one epoch) as in Tab 4 of the paper. Then we finetune on MSR-VTT Retrieval to evaluate the pre-trained model:
>
> | Model             | MSR-VTT R@1 | MSR-VTT R@5 | MSR-VTT R@10 |
> | :--------         | :------:    | :------:    | :--------:   |
> | CLIP-ViP (from scratch)| 28.5       | 55.2        |   68.6       |
> | CLIP-ViP          | 49.6        | 74.5        |   84.8       |
>
> From the results, we can see that there is a huge performance gap. Without basing on pretrained model CLIP, CLIP-VIP can not leverage the rich knowledge entailed in CLIP, thus leading to a much lower training efficiency.

---

### Official Review · Reviewer_P4Rx · 2022-10-24

**Confidence:** 4
**Clarity, Quality, Novelty And Reproducibility:** The quality, clarity and originality …
**Correctness:** 3
**Technical Novelty And Significance:** 3
**Empirical Novelty And Significance:** 3
**Recommendation:** 6

**Strength And Weaknesses:**

Pros:
+ The proposed CLIP-VIP model successfully extends a pre-trained CLIP model to the video-text tasks with only limited revisions, which marginally increases negligible parameters and computations.

+ This paper has systematically explored the crucial factors for model transfer between image-text and video-text tasks, which may shed light on the following works.

+ It has achieved very competitive results on video-retrieval tasks on many benchmarks. Moreover, there are solid ablation analyses for different components as a thorough evaluation.

Cons:
- This paper is not written and organized well. Without extra reading efforts, I cannot easily understand the motivation and the proposed method. Some of the sentences are pretty long and unfriendly to read. The introduction section focuses too much on algorithms/methods instead of its motivation. As a result, I am still confused about the major contributions of this paper.

- The major technical contribution of this paper seems to be proxy video tokens which are simple and trivial. Moreover, how to assign the number of proxy video tokens needs some manual effort, which degenerates its value in practice.

- The post-training is costly, considering the 32 GPUs in use. So, I have not seen the extraordinary benefits of model transfer in efficiency. The authors are suggested to compare the cost of training from scratch and post-training to highlight the advantages of model transfer.

Questions/Other Concerns:

1. How to assign the M (number of proxy video tokens) in practice? What will be if M is small or large?

2. How many GPU hours are needed for post-training? Is it more efficient than other methods?

3. What is the meaning of single frames (F) in page 5? What is the difference between F and video sequences (V)?


**Summary Of The Paper:**

This paper presents a new framework called CLIP-VIP to transfer the image-text model to video-language alignment. It has been found that the scale of dataset is crucial for post-pretraining. Therefore, the authors apply HD-VILA-100M due its large scale and diverse category. The other factor is the language domain gap measured by the NMI score in Table 1. Therefore, CLIP-VIP introduces an additional captioning model (OFA-Caption) to augment text data as another source. To enable cross-model post-pretraining based on video-text pairs, the authors invent video proxy tokens to summarize the content of the whole video temporal-spatially. The model is trained by the info-NCE loss in a self-supervised way.



**Summary Of The Review:**

Based on all the comments above, I am leaning toward weak rejection. I will consider upgrading the score after a strong rebuttal from the authors.

---

> ### Author Response · Authors · 2022-11-18
> **Response to Reviewer P4Rx**
>
> We sincerely thank you for the constructive comments. We will **polish our paper writing, especially the part of motivation and contribution**. We address your major concerns as below.
>
> **1. Technical Contribution**
> To the best of our knowledge, there is no method similar to the proposed video proxy attention mechanism. In our work, the video proxy token acts as a light but effective bridge between image and video. As shown in Tab 2, the proxy-guided attention outperforms Fully-Attention and seqTransformer. Meanwhile, the simplicity greatly increases the extensibility of CLIP-ViP.
>
> Assigning the number of proxy video tokens is not needed for downstream tasks. We first evaluate the token number of 2, 4, 8 on MSR-VTT retrieval and they all lead to a performance boost (Tab 2). Then we choose the best $M=4$ for post-pretraining. The token number is **fixed** during and after the post-pretraining stage.
>
> Moreover, we emphasize our technical contribution including **video proxy** mechanism and **omnisource cross-modal learning** which enables simultaneously processing videos and images in contrastive learning.
>
> **2. Efficiency of Post-pretraining**
> In light of the good image-language representation learned by CLIP, our post-pretraining significantly reduces the training cost. Compared to HD-VILA which trains video-language representation from scratch, **CLIP-ViP outperforms HD-VILA by a large margin (refer to Tab 5,6) with about 1/30 training time**. To make more fair comparison with training from scratch, we add an experiment setting to pre-train from scratch with the same setting and training time (one epoch) as in Tab 4 of the paper. Then we finetune on MSR-VTT Retrieval to evaluate the pre-trained model:
>
> | Model             | MSR-VTT R@1 | MSR-VTT R@5 | MSR-VTT R@10 |
> | :--------         | :------:    | :------:    | :--------:   |
> | CLIP-ViP (from scratch)| 28.5       | 55.2       |   68.6      |
> | CLIP-ViP          | 49.6        | 74.5        |   84.8       |
>
> From the results, we can see that there is a huge performance gap. Without basing on pretrained model CLIP, CLIP-VIP can not leverage the rich knowledge entailed in CLIP, thus leading to a much lower training efficiency.
>
> **3. Other Details**
> * Refer to our response in the first question, the token number is **fixed** after the pre-training stage. The impacts of $M$ is studied in Tab 2. We evaluate on MSR-VTT retrieval to save computation source.
> * Training CLIP-ViP with one epoch takes 2 days on 32 V100. The results shown in Tab 3,4 are under the one epoch setting. Compared with HD-VILA which trains from scratch, CLIP-ViP outperforms HD-VILA by a large margin with about 1/30 training time.
> * Refer to Sec 4.1 in the paper, $F$ is the middle frame of each video in HD-VILA-100M. Our auxiliary captions are extracted on the middle frame.

---

> > ### Comment · Reviewer_P4Rx · 2022-11-26
> > **Post-response Feedback**
> >
> > Dear authors,
> >
> > I really appreciate the efforts of the authors during the rebuttal. But it seems there is no revision of the introduction section in the latest manuscript. By the way, it will be good to highlight the changes in other colors to let reviewers easily notice them. I acknowledge the contribution of proxy tokens, which is interesting. As for efficiency, I am still trying to be convinced since the authors only mentioned the rough comparison with HD-VILA and did not provide the details such as GPU Hours and GPU tapes (if using the similar GPUs to compare). Moreover, how about comparing it with other baseline methods in efficiency apart from HD-VILA? I will consider upgrading the score if the authors have solid proof of efficiency, which seems to be one of the core contributions of this paper.
> >
> > Thanks,
> > Reviewer P4Rx

---

> > > ### Author Response · Authors · 2022-11-28
> > > **Response to Reviewer P4Rx**
> > >
> > > Thanks for your recognition of proxy token's contribution. Thanks for the advice on paper writing. According to the policy, we can not update the draft now. We will polish the final version in Camera Ready. To demonstrate the efficiency of CLIP-ViP, we list the comparison of training cost between CLIP-ViP (pretraining for one epoch) and some strong video-language pre-training baselines here:
> > >
> > >
> > > | Model | PT Data | PT GPU | PT Cost | MSR-VTT R@1 |
> > > | -------- | -------- | -------- | -------- | :--------: |
> > > | Frozen   | CC3M+WebVid2.5M  | A40  | 1.3K GPU Hours |  31.0   |
> > > | HD-VILA  | HD-VILA-100M | V100 | 65K GPU Hours | 35.6 |
> > > | BridgeFormer | CC3M+WebVid2.5M|  A100 | 1.0K GPU Hours | 37.6 |
> > > | All-in-One | Howto100M+WebVid2.5M|  A100 | 5.4K GPU Hours | 37.9 |
> > > | CLIP-ViP (from scratch) | HD-VILA-100M | V100 | 2.0K GPU Hours | 28.5 |
> > > | CLIP-ViP | HD-VILA-100M | V100 | 2.0K GPU Hours | 49.6 |
> > >
> > >
> > > PT Data, PT GPU, PT Cost respectively indicate the data, GPU type and GPU nums * hours during the pre-training stage. Note that A40's performance is slightly better than V100, and the performance of A100 is about twice that of V100, theoretically.
> > >
> > > The results demonstrate that CLIP-ViP achieves much better performance with low training cost, by leveraging the rich knowledge entailed in CLIP. It's worth noting that we choose HD-VILA-100M to benefit from its large scale and diverse sementics, as analysized in Sec. 3.1.

---

> > > > ### Comment · Reviewer_P4Rx · 2022-11-29
> > > > **Post-discussion**
> > > >
> > > > Dear Authors,
> > > >
> > > > Thank you for your hard work in the computation cost comparison. Currently, I am positive about this submission now and will raise my score.
> > > >
> > > > Best,
> > > > Reviewer P4Rx

---

### Official Review · Reviewer_39Ux · 2022-10-25

**Confidence:** 2
**Correctness:** 3
**Technical Novelty And Significance:** 3
**Empirical Novelty And Significance:** 3
**Recommendation:** 6

**Clarity, Quality, Novelty And Reproducibility:**

The quality, clarity, and originality of the work meet the requirement of an ICLR publication.


**Strength And Weaknesses:**

Strength:
The paper is well organized.
The performance is very impressive.


Weaknesses:

1. One primary concern: is the domain gap between language sources really the underlying problem? Do the authors really solve this problem in this work?

1.1) In Table 1, we can see $HD-VILA_{cap}$ has a far smaller NMI than $HD-VILA_{sub}$. If the domain gap is the real problem and $HD-VILA_{cap}$ has the same amount of data as $HD-VILA_{sub}$, in Table 4, $HD-VILA_{cap}$ should have much better performance than $HD-VILA_{sub}$, but we can this is not the truth. Indeed, $HD-VILA_{cap}$ has worse performance than $HD-VILA_{sub}$, and $HD-VILA_{cap}$ is even worse than w/o post-pretrain. This violates the domain gap assumption of the authors.

1.2) In Table 1, CC12M has a smaller NMI than $HD-VILA_{sub}$. In Table 4, we can see only ImageCaption pre-training does not help fine-tuning. There are two possibilities: a) the domain gap assumption does not hold; b) we should also consider the domain gap of visual signals. It is also weird why $HD-VILA_{sub}$ + ImageCaption get such good performance as the two work ordinarily without each other.

1.3) Following 1.2, could the authors provide analyses of domain gaps of visual signals across different datasets?

1.4) I think there is a publication that can explain the improvement in performance when using $HD-VILA_{sub}$ + $HD-VILA_{cap}$ [1]. I purely consider $HD-VILA_{cap}$ as a bootstrapping of captions or data augmentations. Hope the authors can turn my mind around. Besides, I think one way to verify this idea is by adding HowTo100M + HD_VILA result in Table 4. If this does not work, I may lean toward the domain gap assumption.

2. What is the difference between video proxy tokens and video prompts? I think they are just the same thing with various names.


[1] BLIP: Bootstrapping Language-Image Pre-training for Unified Vision-Language Understanding and Generation. https://arxiv.org/abs/2201.12086



**Summary Of The Paper:**

This paper attributes the weakness of post-pretraining of text-video models (based on the text-image model) to the domain gap between video subtitles and image captions. The authors develop several methods to improve the effectiveness of post-pretraining: data augmentations via video captions, video proxy tokens, and omnisource cross-modal learning.


**Summary Of The Review:**

The paper does an excellent work in the post-pretraining of text-video models. I hope the authors can resolve my concern about the domain gap assumption.

---

> ### Author Response · Authors · 2022-11-18
> **Response to Reviewer 39Ux**
>
> We sincerely thank you for the constructive comments. We address your major concerns as below.
>
> **1. Domain Gap problem**
> 1.1) We take advantage of the complementary of subtitles and generated captions. Subtitles are translated texts from audio, usually weakly related to describe the corresponding video. They have different forms and visual correspondence from descriptive texts like captions or web-collected alt-texts. However, they contain rich semantics and dynamic information, and the proportion can be easily scaled up, which are both important for video representation learning. As the language of most video-language tasks is descriptive, we utilize generated captions as auxiliary data.
>
> However, generated captions are **pseudo labels** and usually have **lower diversity** and **limited mode** determined by the captioning model. Therefore, even though they have small domain gap with downstream tasks in terms of language format, the information of generated captions is not enough to support the post-pretraining on large-scale video data. Thus, large-scale real-world data (e.g., video-subtitles in HD-VILA-100M) is still needed in training process to provide rich and diverse supervision.
>
> 1.2) Image-caption pairs **lack temporal information**, which is required by video tasks. And existing caption data has much smaller data scale than CLIP's pretraining data (less than 1/20). This will also lead to the "overfitting" (knowledge forgetting) issue as we analyze in Sec 3. Similar to 1.1), the combination provides rich supervision from large-scale video-subtitle pairs and shrinks the domain gap by image-caption pairs' assistance.
>
>
> 1.3) Different with language's unified form (a sequence of words), for visual signals, video contains the extra dimension of temporality compared with image. Thus, it's difficult to measure the domain gap between these two kinds of signals with different forms.
>
> We also would like to emphasize that we do not aim to study the domain gap of uni-modal information. Instead, we study cross-modal alignment by estimating the domain gap of vision-aware language. The CLIP's text feature extractor is trained to align with visual information, and it entails the key or salient visual information. As a result, the CLIP text features can also reflect information of visual correspondence.
>
> 1.4) Based on 1.1) and 1.2), image captions only work well when acting as auxiliary data. We do not aim to expand data but aim to alleviate the domain gap issue. BLIP's generating and filtering method is hard to be transferred to video domain. The reason is that video subtitles have a totally different form from descriptive text, as they are translated texts from audio, and usually weakly related to the corresponding video. As a result, directly using subtitles for post-pretraining will hinder the cross-modal representation learned from image and descriptive text pairs in CLIP, which is the main problem we tackle in this work.
>
> HD-VILA-100M already contains a large amount of how-to domain videos (about 8M). Adding HowTo100M videos will break the ability to generalize to open-domain, which is verified in HD-VILA's paper.
>
> **2. Video Proxy**
> To the best of our knowledge, existing prompting methods aim to finetune a large model by adding a few words or learnable vectors to the **text** encoder. These prompt vectors act as a prefix of the input sentence to **prompt** the model without modifying the structure. In our work, proxy tokens work as a **transfer** station between global representation and frame features in the **video** encoder. This is guaranteed by the proxy-guided attention mechanism. We have totally different motivation and method from existing prompting models. If our understanding of "video prompts" is not the same as mentioned by the reviewer, we are happy to discuss more.

---

### Official Review · Reviewer_qsPb · 2022-10-25

**Confidence:** 4
**Correctness:** 4
**Technical Novelty And Significance:** 4
**Empirical Novelty And Significance:** 4
**Recommendation:** 8

**Clarity, Quality, Novelty And Reproducibility:**

The paper is clearly written and of high quality. There is good novelty, and I believe the work could be easily reproduced.

**Strength And Weaknesses:**

Strengths:
+ New adaptation of ViT for video-text alignment
+ Captioning strategy for full videos using middle frame.
+ Info-NCE adaptation to include video subtitles and video caption.
+ Outperforms baselines and informative ablations


Questions/comments:
- How are negatives chosen?
There are two different text inputs being contrasted, subtitles and captions. I may have missed it from the text, but I couldn't find any details on the strategies used for getting the negatives for Info-NCE. Could the authors provide these details?


- Using DSL posprocessing on all baselines?
The best reported results in this paper are after using the DSL posprocessing. The author mention that they use this for fairness against other baselines. However, I am not sure if this was used in all the baselines. Can the authors comment on whether this makes sense doing or not?

**Summary Of The Paper:**

This paper proposes an adaptation of the ViT based CLIP model for video-text alignment that learns the spatio-temporal structure of the videos during the alignment. The authors start by analysis the problems with current approaches that straight forwardly apply CLIP on video for video-text alignment. They find that data scale and domain gap are big factors in the performance of models. Based on these findings, the authors design a video proxy to allow the model to understand spatio-temporal aspects of video while also combined captioning and subtitle language information from videos for alignment. In the experiments, the authors show multiple ablations to validate the different components of their method and also outperform that state of the art in many datasets.

**Summary Of The Review:**

All-in-all, I like this paper and I think it should be accepted in the current form. There is analysis of problems from previous methods while tackling this problem. There is novelty in the proposed method and objective strategies, and the method outperforms previous baselines.

---

> ### Author Response · Authors · 2022-11-18
> **Response to Reviewer qsPb**
>
> We sincerely thank you for the recognition and constructive comments on our work. We address your major concerns as below.
>
> **1. Negatives Selection**
> The selection of negatives is described in Sec. 4.3 and illustrated in Fig. 2. Losses (a-d) are contrastive losses using multi-sources. Losses (a-c) select negative samples from either subtitles or captions. Loss (d) expands the negative sample by combining subtitles and captions, which basically doubles the number of negative samples compared with (a-c).
>
> **2. Post-processing Issues**
> As explained in captions of Tab 5, 6, methods with "\*" indicate using post-processing DSL and "$\dagger$" indicate a similar post-processing method QB-Norm. In all the tables, methods without "\*" or "$\dagger$" have no post-processing. By fair comparison, our model without post-processing also achieves the best results compared with other models without post-processing.

---

> > ### Comment · Reviewer_qsPb · 2022-11-29
> > **Response to Reviewer qsPb**
> >
> > Sorry for the late response. I would like to thank the authors for their rebuttal. They addressed all my questions. One thing I would ask is for the authors to make the parts where I had the questions more obvious so it's easier to notice.

---

> > > ### Author Response · Authors · 2022-11-30
> > > **Response to Reviewer qsPb**
> > >
> > > Thanks for the recognition. We will revise these parts in the final version.

---

### Official Review · Reviewer_XjSo · 2022-10-31

**Confidence:** 4
**Correctness:** 3
**Technical Novelty And Significance:** 3
**Empirical Novelty And Significance:** 3
**Recommendation:** 8

**Clarity, Quality, Novelty And Reproducibility:**

* The method includes new components for video-language training from CLIP-like models such as video proxies and proper attention masks, and it has moderate novelty.

* Experimental results are convincing and show superior performance compared to the baselines.

* Presentation is clear and well-motivated.


**Strength And Weaknesses:**

**Strengths**:

* The paper is well-written and the proposed approach is described clearly.

* The authors motivate the problem well with the experiments on the impact of data scale and language domain gap.

* CLIP-ViP outperforms baselines on various datasets such as MSR-VTT, DiDeMo and ActivityNet.

**Weaknesses**:

* The authors only evaluate their approach on the text-to-video retrieval task. It is unclear whether the method also provide benefits in other tasks such as video action recognition. Given that we can formulate video classification as video-text alignment (using text prompts), does the method perform well on this task as well?

* The authors argue that there is a discrepancy between language data in pre-training and downstream tasks, and propose their captioning method accordingly to bridge this gap. As the downstream task is limited to text-to-video retrieval, it is questionable how generalizable the captioning approach is. Providing results on other downstream tasks can clarify this.

* Using the additional captioning model adds extra compute to the process that needs to be discussed.

* Regarding the plots in Figure 1, does this overfitting hold for different values of learning rate and optimization schedule?

* There are a few errors in the manuscript, e.g. section 4.3. "we definite" -> "we define"

**Summary Of The Paper:**

The paper proposes a method for adapting image-text pre-trained models to video-language alignment tasks (specifically text-to-video retrieval). It argues there is a domain gap between text used in video-language pre-training versus downstream tasks, and uses an image captioning model to caption middle frames of the video and use them alongside with the subtitle text in training. It also adds additional global video tokens to the visual encoder and uses an attention mask to mask attention from tokens of each frame to those of other frames. Experiments on text-to-video retrieval show that the proposed approach outperforms competing methods.

**Summary Of The Review:**

Overall, the proposed approach is sound and shows reasonable performance gain in the experiments. However, it is mainly evaluated on a single task which makes it questionable to what extent the method is generalizable to other tasks.

---

> ### Author Response · Authors · 2022-11-18
> **Response to Reviewer XjSo Part 2**
>
>
> **2. Additional Computation of Captioning**
> The goal of pre-training is to provide downstream tasks with powerful models. So it is common to add auxiliary information or additional computation during pre-training stage. For example, OSCAR [3] uses an off-the-shelf object detector to include more information for input and provide supervision for pre-training. MVP, BEiT V2, and BEiT-3 [4,5,6] use CLIP as supervision in pre-training. And subtitles in video dataset are extracted by Automatic Speech Recognition (ASR) models. In our paper, we only involve captioning model in the auxiliary captions generation. This **offline** process can be regarded as a **data preparation** for our pre-training. **There is no computation of the captioning model during pre-training and training/testing on downstream tasks.** The generated captions will also be released to benefit future works in a once-for-all manner.
>
> **3. Experiments in Preliminary Analysis**
> For Fig 1's experiment, We try various hyper-parameters for training, including small learning rates or frozen encoders. The conclusion stays the same. It's worth noting that the "overfitting" here has the different meaning. Usually overfitting means the model fits too well on its training data, thus cannot perform accurately against unseen data. However, in our scenario, "overfitting" means further training on new data will cause degradation of the knowledge learned by CLIP model. Due to the difference, this phenomenon cannot be easily solved by optimization tricks.
>
>
> >[1] Kuehne, Hildegard, et al. "HMDB: a large video database for human motion recognition." ICCV 2011.
>
> >[2] Goyal, Raghav, et al. "The" something something" video database for learning and evaluating visual common sense." ICCV 2017.
>
> >[3] Li, Xiujun, et al. "Oscar: Object-semantics aligned pre-training for vision-language tasks." ECCV 2020.
>
> >[4] Wei, Longhui, et al. "MVP: Multimodality-guided Visual Pre-training." arXiv preprint arXiv:2203.05175 (2022).
>
> >[5] Peng, Zhiliang, et al. "Beit v2: Masked image modeling with vector-quantized visual tokenizers." arXiv preprint arXiv:2208.06366 (2022).
>
> >[6] Wang, Wenhui, et al. "Image as a foreign language: Beit pretraining for all vision and vision-language tasks." arXiv preprint arXiv:2208.10442 (2022).
>
> >[7] Wang, Rui, et al. "Bevt: Bert pretraining of video transformers." CVPR 2022.

---

> ### Author Response · Authors · 2022-11-18
> **Response to Reviewer XjSo Part 1**
>
> We sincerely thank you for the constructive comments. We have polished our main paper or the appendix per your advices. We address your major concerns as below.
>
> **1. More Downstream Tasks**
> We add various suitable downstream tasks to verify that our model can be generalizable to other tasks. Specifically, we add four zero-shot evaluations on video action recognition and video-language multi-choice tasks:
>
> * HMDB51 Action Recognition [1]. HMDB51 is a video dataset containing realistic videos from various sources, including movies and web videos. There are 51 action categories (such as “jump”, “kiss” and “laugh”). We evaluate our model by zero-shot classification on HMDB51. To adapt to our model, we formulate video classification as video-text alignment, using text prompts "a person [label]". The prediction is made on top-1 similarity between query video and prompted texts. We directly test our model without finetuning to report zero-shot accuracy results.
>
> * SomethingSomethingV2 (SSv2) [2] Action Recognition. SomethingSomethingV2 dataset is a large collection of labeled video clips that show humans performing pre-defined basic actions with everyday objects. It requires models to develop fine-grained understanding of basic actions that occur in the physical world. To adapt to our model, we design a multi-choice task on SSv2. The model has to choose the ground truth annotation among all categories. We directly test our model without finetuning to report zero-shot accuracy results.
>
> * MSR-VTT Multi-Choice (MC). MSR-VTT MC is a benchmark built on MSR-VTT video dataset. Given a video query and five descriptive sentences, MSR-VTT MC requires the model to choose the single best answer in the candidates. We directly test our model without finetuning to report zero-shot accuracy results.
>
> * LSMDC Multi-Choice (MC). LSMDC MC is a benchmark built on movie data. Similar with MSR-VTT MC, it requires the model to choose the single best answer from 5 candidates. We report zero-shot accuracy on 10000 test videos.
>
>
> | Model          | HMDB51   | SSv2     | MSR-VTT MC | LSMDC MC |
> | :--------      | :------: | :------: | :--------: | :--------: |
> | CLIP (meanPool)| 40.3     | 25.8     |   89.9     |     64.3      |
> | CLIP-ViP w/o post-pretraining     | 41.1     | 26.1     |   89.4     |   63.5     |
> | CLIP-ViP w/ video-subtitle| 36.6| 28.0     |   86.6     |     67.0        |
> | CLIP-ViP full| **44.8**| **33.0**     |   **90.3**     |    **67.6**      |
>
>
> From the results, we can find that post-pretraining CLIP on subtitle alone will lead to a performance drop on some tasks: HMDB51 and MSR-VTT MC. This phenomenon is in consistency with our assumption in the paper that subtitles usually have unique form compared to captions. By our full setting, CLIP-ViP achieves better results than CLIP on all evaluations, especially on SSv2 which requires more understanding on temporality [7]. These results demonstrate that CLIP-ViP outperforms CLIP on various video benchmarks and further show the generalization of our model CLIP-ViP.

---

### Comment · Reviewer_qsPb · 2022-11-18
**No response from authors?**

It looks like authors did not provide a rebuttal or they decided to make their response to other reviewers only visible by those reviewers. Is this the case?

---

> ### Author Response · Authors · 2022-11-18
> **We will post the response before the deadline of Nov. 18th**
>
> Dear Reviewer qsPb, we are still drafting the rebuttal and revising the paper. We will post the response before the deadline, which could be very soon. Thanks for your patience.

---

> > ### Comment · Reviewer_qsPb · 2022-11-18
> > **RE: No response from authors?**
> >
> > Thanks for your quick response. It would have been good if there was time for a discussion between the authors and reviewers, but I hope that there is no need for this given your rebuttal.

---

> > > ### Author Response · Authors · 2022-11-18
> > > **RE: RE: No response from authors?**
> > >
> > > I think we can still answer the reviewers' questions in Discussion Stage 2 by Dec. 12 per request. Yes, we hope our rebuttal can resolve all reviewers' concerns.

---

### Public Comment · ~Xiao_Wang14 · 2023-03-27
**Could you discuss these two baseline methods of Video Proxy Mechanism?**

On the proposed video proxy mechanism, it is worth mentioning that similar mechanisms have been proposed by some papers using different names, such as "Memory tokens"[1] and "Cross-frame communication transformer"[2].  The only difference is that their proxy tokens can only see patch embeddings within frames.

Therefore, I think it would be better to compare them.


[1] Sukjun Hwang, et al. "Video Instance Segmentation using Inter-Frame Communication Transformers." Neurips 2021.
[2] Bolin Ni, et al. "Expanding Language-Image Pretrained Models for General Video Recognition." ECCV 2022.

---

### Decision · Program_Chairs · 2023-01-20

**Decision:**

Accept: poster

**Justification For Why Not Higher Score:**

The model has strong performance against relevant benchmarks, but all reviewers mark that the overall novelty of the paper as "Aspects of the contributions exist in prior work." A higher score would also require more detailed and insightful ablations, specifically on the existence and effect of the "domain gap", which is a central part of the motivation of the work. Two (of five) reviewers cite this as their main concern, with one still recommending acceptance.

**Justification For Why Not Lower Score:**

The authors showed strong performance on the video retrieval task. Reviewers appreciate the quality of the authors' feedback and the preparation of an updated paper. Only one reviewer (out of 5) does not explicitly recommend acceptance of the paper on the basis that the computational complexity needs to be discussed more. However, authors provide analysis responsive to the reviewer's comments, which the AC thinks should address the concern.


**Metareview: Summary, Strengths And Weaknesses:**

Summary

This paper studies how to best apply the image-based CLIP model to video-text tasks, and how to close the domain gap between text used in image-language pre-training versus downstream tasks, specifically text-to-video retrieval. Authors find that amount of data is a major factor. The paper uses the image captioning model to caption middle frames of the video and use them together with "global" video tokens alongside with the subtitle text in cross-modal multi-source training. The spatio-temporal structure of video is learned with a masking structure and alignment, and the authors show multiple ablations to validate the different components of their method. They outperform the state of the art on many datasets.

Strengths

- The paper starts from a well-defined scientific question, and provides ablations that support the authors hypotheses and conclusions. The vast majority of reviewer questions were answered during the discussion and taken into account. Results are found to be "convincing"
- The paper is well-written and the proposed approach is described clearly. One reviewer finds that the paper contains long sentences, but the overall consensus is that the paper is well written.
- CLIP-ViP outperforms baselines and comparable work on various relevant datasets such as MSR-VTT, DiDeMo and ActivityNet.
- The paper presents a new adaptation of ViT for video-text alignment, a captioning strategy for full videos using middle frame, and adapts Info-NCE loss to include video subtitles and video caption. It outperforms baselines and informative ablations

Weaknesses

- The authors only evaluate their approach on the text-to-video retrieval task. It is unclear whether the method also provide benefits in other tasks such as video action recognition.
- The authors argue that there is a discrepancy between language data in pre-training and downstream tasks, and propose their captioning method is able to bridge this gap. As the downstream task is limited to text-to-video retrieval, it is not clear how geheralizeable the method is
- The method can become computationally expensive, and ablations on pre-training would be useful, as they would help reproducibility.




**Note From Pc:**

if the above contains the word "oral" or "spotlight" please see: "oral" presentation means -> notable-top-5% and "spotlight" means -> notable-top-25%. As stated in our emails, we are disassociating presentation type from AC recommendations

**Summary Of Ac-Reviewer Meeting:**

n/a